# The Effect of Higher Level Computerized Clinical Decision Support Systems on Oncology Care: A Systematic Review

**DOI:** 10.3390/cancers12041032

**Published:** 2020-04-22

**Authors:** Sosse E. Klarenbeek, Harm H.A. Weekenstroo, J.P. Michiel Sedelaar, Jurgen J. Fütterer, Mathias Prokop, Marcia Tummers

**Affiliations:** 1Department of Radiology, Nuclear Medicine and Anatomy, Radboud Institute for Health Sciences, Radboud University Medical Center, 6500 HB Nijmegen, The Netherlands; 2Department of Urology, Radboud Institute for Health Science, Radboud University Medical Center, 6500 HB Nijmegen, The Netherlands; 3Department for Health Evidence, Radboud Institute for Health Sciences, Radboud University Medical Center, 6500 HB Nijmegen, The Netherlands

**Keywords:** clinical decision support system, neoplasm, systematic review, implementation, evidence-based medicine

## Abstract

Background: To deal with complexity in cancer care, computerized clinical decision support systems (CDSSs) are developed to support quality of care and improve decision-making. We performed a systematic review to explore the value of CDSSs using automated clinical guidelines, Artificial Intelligence, datamining or statistical methods (higher level CDSSs) on the quality of care in oncology. Materials and Methods: The search strategy combined synonyms for ‘CDSS’ and ‘cancer.’ Pubmed, Embase, The Cochrane Library, Institute of Electrical and Electronics Engineers, Association of Computing Machinery digital library and Web of Science were systematically searched from January 2000 to December 2019. Included studies evaluated the impact of higher level CDSSs on process outcomes, guideline adherence and clinical outcomes. Results: 11,397 studies were selected for screening, after which 61 full-text articles were assessed for eligibility. Finally, nine studies were included in the final analysis with a total population size of 7985 patients. Types of cancer included breast cancer (63.1%), lung cancer (27.8%), prostate cancer (4.1%), colorectal cancer (3.1%) and other cancer types (1.9%). The included studies demonstrated significant improvements of higher level CDSSs on process outcomes and guideline adherence across diverse settings in oncology. No significant differences were reported for clinical outcomes. Conclusion: Higher level CDSSs seem to improve process outcomes and guidelines adherence but not clinical outcomes. It should be noticed that the included studies primarily focused on breast and lung cancer. To further explore the impact of higher level CDSSs on quality of care, high-quality research is required.

## 1. Introduction

With annually >20 million new cancer cases expected worldwide by 2025, cancer is a major health problem [1]. Aging of populations in most economically developed countries challenges care by increasing the cancer rate and multimorbidity [2]. Consequently, new diagnostic and therapeutic options for cancer care are developed, tested and implemented at an accelerated pace [3]. Reading and appropriately weighing these options while incorporating patient characteristics has become a task of increasing difficulty for most clinicians [3]. As a result, decision-making based on clinicians’ personal experience and preference takes precedence, resulting in an unwanted variability in quality of care [4].

In response to this growing complexity in cancer care, computerized clinical decision support systems (CDSSs) are developed. These systems aim to support high quality of care and improve decision-making [5,6]. Usually, the basis of these knowledge systems is formed by combining information obtained from electronic health records (EHR) with clinical practice guidelines. Following the article of O’Sullivan et al., CDSSs can be distinguished in simple, mid- and complex level systems [7]. He defines simple level CDSSs as CDSSs that are usually embedded in order entry systems and verify whether the input provided by a clinician is allowable or within a specified range. The output of the CDSS is often an alert or reminder. Mid-level CDSSs comprise of prognostic calculators and automated clinical guideline systems, whereas complex level CDSSs use artificial intelligence, data mining or statistical methods to generate patient specific recommendations [7]. In this review we will focus on mid- and complex level systems as described by O’Sullivan et al., using the term higher level CDSSs.

Despite identification of factors that contributed to the success of CDSSs in clinical healthcare by several reviews in the past years [8,9,10], little is known about the impact of CDSSs on quality of care in oncology. A review of Pawloski et al. suggested that these systems positively impact the quality of cancer care delivered [11]. However, looking at the definitions used by O’Sullivan et al. [7], the CDSSs included in this review could be interpreted as simple level systems. To expand to this work, we performed a systematic review to explore the impact of higher level CDSSs on quality of care in oncology, operationalized in terms of process outcomes, guideline adherence and clinical outcomes.

## 2. Materials and Methods

### 2.1. Protocol and Registration

The protocol for this systematic review is registered in the PROSPERO database (CRD42019124800) [12].

### 2.2. Literature Search

To evaluate the impact of higher level CDSSs on quality of oncology care, we performed a systematic review following the Preferred Reporting Items for Systematic Reviews and Meta Analysis Protocols 2015 (PRISMA-P 2015) [13]. A librarian was consulted for the search strategy; the conducted search combinations are shown in Appendix A. The search strategy combined synonyms for ‘CDSS’ and ‘cancer.’ Pubmed, Embase, The Cochrane Library, Institute of Electrical and Electronics Engineers (IEEE), Association of Computing Machinery (ACM) digital library and Web of Science were systematically searched. Time limits were from January 1, 2000 to December 31, 2019. In addition, reference lists of systematic reviews identified during screening on title and abstract were searched for relevant papers [14,15,16,17,18,19,20].

### 2.3. Data Collection

The retrieved articles were imported into EndNote and duplicates were removed. First, titles and abstracts of articles resulting from the search strings were screened independently by two researchers (S.K. and H.W.). Studies meeting the inclusion criteria, as stated below, were selected for full text screening. Second, the full text papers were retrieved and reviewed (S.K. and H.W.). After the screening process, the inter-rater agreement was calculated. Any disparities were discussed at regular intervals and resolved via consensus. If required, a third reviewer was consulted to resolve disagreement (M.T.). The web-based software Rayyan was used throughout the process [21].

### 2.4. Study Selection Criteria

CDSSs considered relevant for this review included higher level systems. This are mid- and complex level as defined by O’Sullivan et al. [7]. Systems others than mid- and complex level were excluded. Furthermore, only CDSSs applied in cancer care or cancer related treatments were considered. CDSSs in a pre-clinical stage were excluded; this was defined by use of the Technology Readiness Level (TRL) analyses and stated as a TRL < 7 [22]. Studies assessing process outcomes, guideline adherence and clinical outcomes were included. Process outcomes are defined as outcomes related to workflow, workload and costs. Adherence to guidelines denotes the degree of compliance between clinicians’ decision or action and the recommendation of clinical guidelines. Clinical outcomes are measurable changes in patients’ health, function or quality of life that result from care. Studies that assessed the effect on clinical outcomes were divided in two groups: (a) studies assessing the effect of less clinical consumption for equivalent clinical outcomes and (b) studies assessing the effect of CDSSs directly on clinical outcomes. No limitations were imposed with regard to language. The study types eligible for inclusion were randomized controlled trials, non-randomized controlled trials and before-and-after studies. Observational studies and studies without control groups were excluded.

### 2.5. Extraction and Assessment of Quality

Data was extracted using the Cochrane Collaboration’s double-data collection and extraction methodology [23]. The following data were retrieved by use of a structured data collection form: study year, study design, cancer type, system classification, system features, clinical topic addressed, outcome parameters, results, sample size, control and intervention group. Summative tables were created with descriptive comparison between the studies in outcomes and study characteristics. Due to heterogeneity of the data, meta-analyses was not possible. Outcomes were categorized in process outcomes, guideline adherence and clinical outcomes and analyzed per category. If studies evaluated more than one outcome category, the same study can be found in multiple tables. In addition, risk of bias for non-randomized studies and quality of evidence were graded by use of the ROBINS-I tool and GRADE scale [24,25].

## 3. Results

### 3.1. Eligible Studies

With the search strategy presented in Appendix A, 13,919 records were identified. After removal of duplicates, 11,397 studies were selected for screening. After screening of title and abstracts, 61 full-text articles were assessed for eligibility and 9 studies were included in the final analysis. The PRISMA-P 2015 flowchart (Figure 1) provides more information about the literature selection process.

The percentage of inter-rater agreement during screening of title and abstract was 98.9% and during screening of full-text 95.1%.

### 3.2. Study Characteristics

Table 1 describes the characteristics of the included studies. All studies were conducted in adults (n = 7985; 78.8% female) in the United States (4/9) or Europe (5/9). Six studies were performed in a single center, two studies in multiple centers and for one study this was not defined in the article. Types of cancer included colorectal cancer (n = 245 individuals; 3.1% of total number of individuals in these studies), lung cancer (n = 2227; 27.8%), prostate cancer (n = 325; 4.1%), breast cancer (n = 5040; 63.1%) and other cancer types (n = 148; 1.9%). Clinical topics that were addressed by use of CDSSs are colony stimulating factor (CSF) support to manage chemotherapy-induced febrile neutropenia (2/9), treatment of tumor-induced pain (1/9), treatment decisions in multidisciplinary team (MDT) meetings (2/9), pain management (1/9), treatment planning (1/9) and follow-up based on carcino-embryonic antigen (CEA) testing (1/9). All studies targeted the physicians as users. Six studies targeted multiple specialisms; the others focused on one specialism. CDSSs of included studies focused on risk assessment or therapy options to support informed decisions [26,27,28,29,30,31,32], as well as statistical methods and data mining to generate patient-specific recommendations [33,34]. For most studies (7/9), clinical guidelines formed the knowledge base of the CDSS.

### 3.3. Impact of CDSS on Oncology Practice

Table 2, Table 3 and Table 4 show an overview of the outcomes measured in the nine studies included in this systematic review.

#### 3.3.1. Process Outcomes

Six studies reported process outcomes (Table 2). The issues addressed included: use of CSF to control chemotherapy-induced febrile neutropenia [26,27], physician-prescribing behavior [32], clinical trial rate [32], attainment of analgesia [28], time to analgesia [28], frequency of pharmaceutic intervention and pain assessment [28], costs [33] and workload [31]. Comparators included usual care, based on either MDT decisions or clinical guidelines.

Implementation of a CDSS was shown to positively impact the following process outcomes:Use of CSF: Febrile neutropenia (FN) frequently complicates cancer chemotherapy; therefore CSF is often administered to reduce the risk and severity of FN. However, 30% of the patients that receive CSF have a low risk of FN, increasing costs and patient burden with no clinical benefit. Agiro et al. and Adeboyeje et al. showed that implementation of CDSSs significantly decreased the use of CSF as primary prevention for FN, which was interpreted as reduced overtreatment [26,27].Physician-prescribing behavior: Bouaud et al. showed that initial decisions in breast cancer management were modified in 31% of cases after implementation of a CDSS. Whatever the motivation for change, it was always directed towards an improvement in patient management [32].Frequency of pain assessment: Christ et al. showed that frequency of nursing pain assessments within 24 h after admission was significantly higher in the postimplementation group compared to the pre-implementation group (12.0 vs. 7.4 *p* < 0.001). Increased frequency of nursing pain assessment is associated with improved pain outcomes [28].Healthcare costs: Jackman et al. suggests that the use of a CDSS resulted in a significant reduction in total costs for stage IV non-small cell lung cancer patients one year after diagnosis, by approximately $17,000 (from $69,122 before to $52, 037) [33].Clinician’s workload: Verberne et al. calculated that on average a clinician needs 64 min per patient per year in the follow-up of colorectal cancer (CRC). Clinicians’ workload was significantly reduced to 23 min per patient per year after implementation of the CDSS, saving more than 40 min per patient per year during follow-up [31].

No significant effects of implementation of a CDSS were found by Christ et al. and Bouaud et al. for the number of patients recruited for clinical trials, attainment of analgesia, time to analgesia and frequency of documented pharmaceutic interventions [28,32].

#### 3.3.2. Guideline Adherence

Five studies assessed adherence to clinical practice guidelines (Table 3) [28,29,30,32,34]. Three studies showed improvement in guideline adherence in breast cancer management [29,30,32], two studies showed conflicting results in guideline adherence in pain management [28,34] and no significant difference in guideline adherence in prostate cancer management was reported [29].

Three studies reported a significant increase in adherence to breast cancer guidelines (Oncolor guidelines, Cancer Est) after implementation of a CDSS ranging from 9.3% to 24.1% [29,30,32]. Rios et al. suggested no significant effect of a CDSS on guideline adherence in prostate cancer management to CancerEst guideline recommendations. 

Bertsche et al. showed a 60% reduction in number of deviations from pain management guidelines (WHO principles) [34]. In contrast, the study of Christ et al. showed no significant difference in compliance rate of pharmacist’s decisions to NCCN guidelines on pain regimes for oncology patients after implementation of a CDSS.

#### 3.3.3. Clinical Outcomes

In total six studies assessed the effect of higher level CDSSs on clinical outcomes (Table 4). 

Four studies assessed the effect of less clinical consumption for equivalent clinical outcomes [26,27,31,33]. Adeboyeje et al. showed a 9% reduction in CSF use in lung cancer management after implementation of a CDSS, while the FN rate did not change significantly [26]. Agiro et al. showed similar results for CSF use in breast cancer management [27]. Jackman et al. reported a significant decrease in the average 1-year costs of care after implementation of a clinical pathway for lung cancer, with no compromise in survival [33]. Last, Verberne et al. showed a reduction in clinicians’ workload after implementation of a CDSS for CRC management, with no significant difference in metastases found in follow-up [31].

Two studies assessed the effect of CDSSs directly on clinical outcomes, neither showed a significant effect of CDSSs on clinical outcomes. Bertsche et al. showed no significant change in pain intensity score (NVAS) on day 5 after admission, after implementation of a CDSS [34]. Similar, Christ et al. showed no significant difference in mean pain score at hospital admission and after 28 h [28].

### 3.4. Risk of Bias and Level of Evidence

With use of the Robins-I tool, the risk of bias is rated critical in six [28,29,30,31,32,34] studies and serious in three studies [26,27,33]. An overview of the risk of bias scoring is provided in Figure 2.

Based on the GRADE scale, the level of evidence had to be rated low in three studies [26,27,33] and as very low in six studies [28,29,30,31,32,34].

## 4. Discussion

### 4.1. Summary of Evidence

We systematically reviewed scientific literature and identified the impact higher level CDSSs can bring to oncology clinical practice, operationalized in terms of process outcomes, guideline adherence and clinical outcomes. Overall, implementation of CDSSs resulted in significant improvements for process outcomes and guidelines adherence but no significant differences for clinical outcomes. However, all studies show a critical or serious risk of bias and a low to very low quality of evidence. While this review suggests benefits of higher level CDSS for oncology practice, there is the need for high-quality research.

### 4.2. Overall Completeness and Applicability of Evidence

The applicability of the evidence is limited due to several reasons. None of the included studies evaluated deviations from the intended CDSS use. However, literature shows that improper use of CDSSs did occur in other studies. Improper use can be defined as misusing the software or not using the software to its full capacity. Whereas proper use can be defined as correctly using the software by accurately processing and interpreting patient specific data [35,36]. An example of improper and proper use is given in previous research by Bouaud et al. in which physicians’ attitudes towards the advice of a decision support system was explored [35]. This study evaluated the way a CDSS (Oncodoc2) was used and the impact of the system on MDT decision compliance with CPGs. Distinguished were proper use, decisions made with correct CPG navigations and improper use, decisions made with incorrect CPG navigations. In case of proper use, MDT navigations were identical to reference navigations and in case of improper use, MDT navigations differed from reference navigations. Reported was that 36.8% of the CPG navigations performed in Oncodoc2 were done improperly and 33.9% properly. The compliance rate was significantly different according to the quality of navigations; 94.2% for correct navigations, 80.0% for incorrect navigations and 90.2% for missing navigations as no decision support system was used [35]. Proper use of CDSSs is of major importance to ensure patient safety and to evaluate the effect of a CDSS. Bouaud et al. concluded that it was better not to use the CDSS than to use it improperly, as their results suggested improper use caused lower decision quality than not using the system [35]. Therefore, without detailed information on deviations from intended use the applicability of evidence is limited.

Moreover, in all studies the CDSSs are practiced in a specific clinical area within the oncology. Due to this, the generalizability of the results to other types of cancer or tests is limited. Changes in system features are probably necessary before results can be extrapolated to other cancer types and workflows. Additionally, except for two, all studies were performed in a single center setting. Further, several included studies lacked transparency in missing data. For example, Rios et al. included 39 patients but only 30 patients are reported in the results [29]. These missing data can result in an under- or overestimation of the reported effect.

Conflicting results were found for adherence to pain management guidelines. Bertsche et al. showed a 60% reduction in deviations from pain management guidelines [34], while Christ et al. found no significant difference in guideline adherence after implementation of a CDSS [28]. These conflicts may be explained by differences in guidelines (WHO principles, NCCN guidelines), study population and definition of guideline adherence.

### 4.3. Agreements and Disagreements with Literature

The majority of studies included in this review evaluated the impact of CDSSs on process outcomes in oncology care. This is in line with the review of Pawloski et al., which assessed the impact of CDSSs in oncology practice and described positive impacts on prescriber errors, safety events and workflow [11].

Outside the scope of oncology, several reviews reported positive effects of CDSSs on process outcomes. The review of Jia et al. assessed the effects of CDSSs on medication prescription, medication dose and other drug related outcomes [37]. This study reported that CDSSs significantly impact medication administration in 108 out 143 studies included [37]. The review of Bright et al. assessed CDSSs with a broad scope in healthcare [38]. Modest evidence is reported from academic and community inpatient and ambulatory settings, showing reduced hospitalization expenses with CDSS use and a positive effect on costs compared with control groups and other non-CDSS intervention groups [38]. No prior research and reviews evaluated the impact of CDSSs on workload. The reduction in clinicians’ workload reported by Verberne et al. should be confirmed in future research [31]. Similar, we found significant effects for drug administration, costs and workload. However, comparison with previous reviews is challenging since our results were reported specifically within oncology care.

Previous reviews demonstrated a positive impact of CDSSs, that can be interpreted as simple level, on guideline adherence in and outside oncology [11,39,40]. These findings are consistent with most of our results. All studies in this review showed a significant increase in guideline adherence, except for one study on CDSS in prostate cancer. The later was explained by Rios et al. as a result of less complex decision-making within prostate cancer management. Overall, results suggest that CDSSs can translate the complex data into clinical meaningful support and could ultimately result in less practice variation.

The effect of CDSSs on clinical outcomes is sparsely studied [8,41,42]. Previous reviews mainly focused on CDSSs that can be interpreted as simple level. They reported only a few studies that have positive benefits on patient outcomes. Lack of clinically important findings may be due to inappropriate study designs for measuring clinical outcomes. Mostly because of limited follow-up times and small sample sizes. These findings are in line with our results; two studies evaluated the direct impact of CDSSs on clinical outcomes and showed no significant differences after implementation of a CDSS [28,34].

### 4.4. Strengths and Limitations

Our systematic review has several strengths. Over 10,000 articles were screened by two independent researchers following the PRISMA-P 2015 and the Cochrane Collaboration’s double-data collection and extraction methodology [13]. We are the first to specifically evaluate the impact of higher level CDSSs. By doing this, we expand the current landscape, complementary to the systematic review of Pawloski et al. [11].

Our review also has limitations. First, higher level CDSS may be misclassified as low level during the screening process due to limited or unclear information. Subsequently, exclusion of misclassified CDSSs cannot be rule out. Second, general observations on higher level CDSSs are limited due to small numbers of studies included and heterogeneity in interventions, populations, settings and outcomes. Third, we acknowledge that publication bias and selective reporting cannot be excluded. Finally, guided by the PRISMA-P 2015 we used the GRADE scale to rate the quality of evidence of the included studies. In the GRADE approach non-randomized studies can provide high quality of evidence but will automatically be downgraded for limitations in design such as lack of blinding [25]. In the evaluation of CDSSs a double blind design is not possible, as caregivers have to use the system. Therefore, even a well-designed RCT evaluating a CDSS will be rated low level of evidence. In conclusion, before discarding the results of CDSS evaluation, the level of evidence has to be taken into consideration carefully.

## 5. Future Research

Higher level CDSSs have the potential to significantly reduce workload, costs and to increase guideline adherence in clinical practice. To achieve this CDSSs need to contain one of the following features—automated CPGs, algorithms that define best fitting care pathways or algorithms based on CPGs. Additionally, CDSSs have to be web-based or integrated in the EMR and be able to manage big data including EMRs, disease registries, patient surveys and information exchanges. If these features are incorporated in higher level CDSSs, multiple facets of patient data can be used to optimize the diagnostic and therapeutic pathways of individual patients.

The reported significant effects of this review are based on studies rated with a low to very low quality of evidence and serious or critical risk of bias. Still, these outcomes demonstrate the potential value CDSSs can bring to oncology practice but also emphasizes the critical need for high quality studies. To ensure this quality, future research should include prospective analysis with appropriate study designs, for example, power analysis, long term follow-up and analytic methods that are necessary to demonstrate evidence-based improvements and to build the case for wider implementation.

## 6. Conclusions

The number of studies that evaluated impact of higher level CDSSs in oncology care is sparse. The reported evidence suggests that higher level CDSSs improve process outcomes and guidelines adherence. So far, no evidence was found for improved clinical outcomes. To further explore the impact of higher level CDSSs on the quality of care, high-quality research for reliable evidence is required.

## Figures and Tables

**Figure 1 cancers-12-01032-f001:**
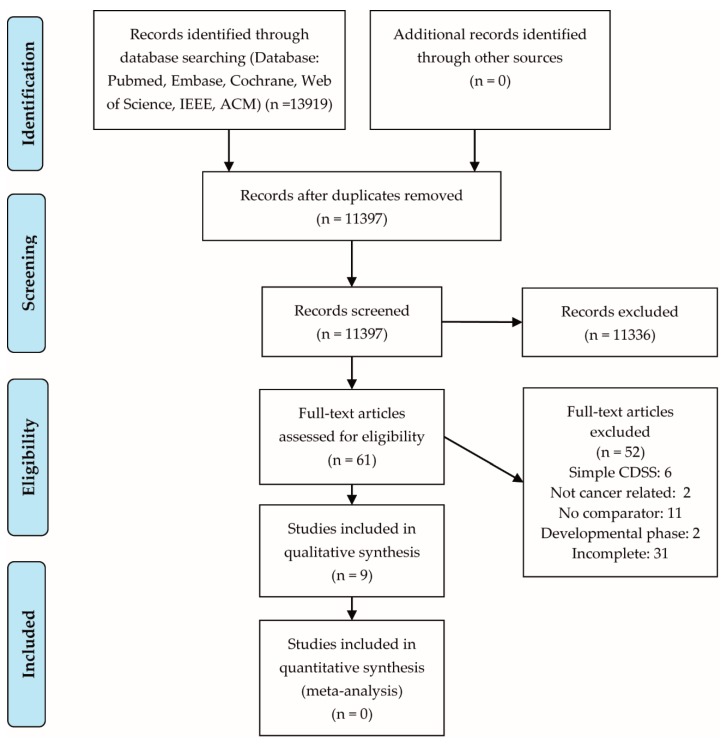
PRISMA-P 2015 flowchart.

**Figure 2 cancers-12-01032-f002:**
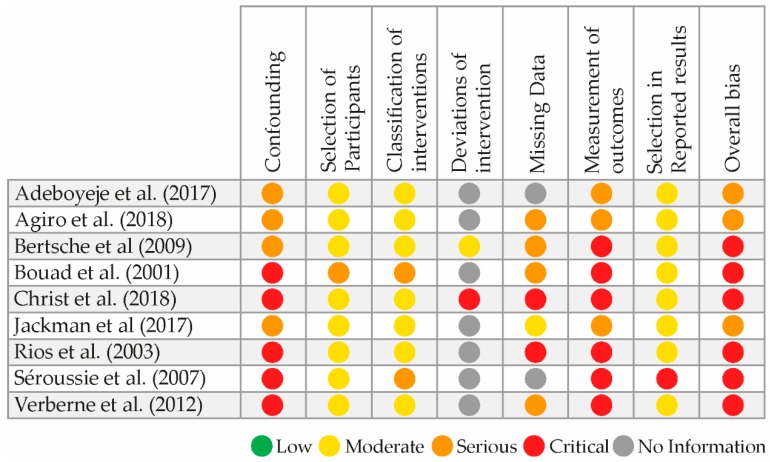
Risk of bias assessment.

**Table 1 cancers-12-01032-t001:** Study Characteristics.

Study	Study Design	Cancer	System Classification	Clinical Topic	Outcome Parameters	Total Sample Size	Control (*N =* Number of Participants)	Intervention (*N =* Number of Participants)	Risk of Bias	Quality of Evidence
Adeboyeje et al., 2017 [26]	Multicenter, before-after, cohort study	Lung	Decision support system	CSF support for chemotherapy	(1) % CSF use (2) % at high risk for febrile neutropenia	1857	National guideline (*N =* 707)	CDSS (*N =* 1150)	Serious	Low
Agiro et al., 2018 [27]	Multicenter, before-after, cohort study	Breast	Decision support system	CSF support for chemotherapy	(1) % CSF use (2) % at high risk for febrile neutropenia	4001	National guideline (*N =* 1991)	CDSS (*N* = 2010)	Serious	Low
Christ et al., 2018 [28]	Single center, before-after, cohort study	Hematologic malignancies and solid tumors	Decision support system	Pain management of opioid-tolerant oncology patients	(1) Attainment of analgesia (2) Frequency medication (3) Frequency pain assessment (4) Guideline adherence (5) Pain score	62	National guidelines (*N =* 30)	CDSS (*N =* 32)	Critical	Very low
Rios et al., 2003 [29]	Before-after, cohort study *	Breast and prostate	CPG system	Treatment planning	Guideline adherence	907	Standard of care (*Breast N =* 320, *prostate N =* 188)	CDSS (*Breast N =* 270, *prostate N =* 129)	Critical	Very low
Seroussi et al., 2007 [30]	Single center, before-after, cohort study	Breast	Decision support system	Treatment decisions by MDT	Guideline adherence	316	MDT (*N =* 139)	MDT supported by CDSS (*N =* 177)	Critical	Very low
Verberne et al., 2012 [31]	Single center, before-after, cohort study	Colorectal	Decision support system	Follow-up based on CEA testing	(1) Workload clinicians for follow-up (2) % of metastases found at follow-up (3) % curative metastasectomy	245	Standard of care (*N =* 61)	CDSS (*N =* 184)	Critical	Very low
Bouaud et al., 2001 [32]	Single center, before-after, cohort study	Breast	CPG system	Treatment decisions by MDT	(1) Treatment decision (2) Clinical trial inclusion rate (3) Compliance to CPG	127	MDT (*N =* 127)	MDT supported by CDSS (*N =* 127)	Critical	Very low
Jackman et al., 2017 [33]	Single center, before-after, cohort study	Non-small cell lung	Clinical pathway	Treatment for stage IV	(1) Costs (2) Overall survival	370	Standard of care (*N =* 160)	CDSS (*N =* 210)	Serious	Low
Bertsche et al., 2009 [34]	Single center, before-after, cohort study	All types	Decision support system	Treatment of tumor-induced-pain	(1) % deviation from guideline (2) Pain score	100	Standard of care (*N =* 50)	CDSS (*N =* 50)	Critical	Very low

CSF = colony stimulating factor, CDSS = computerized clinical decision support system, CPG = clinical practice guidelines, MDT = multidisciplinary team, CEA = blood biomarker Carcino-Embryonic Antigen, * Number of centers is not mentioned.

**Table 2 cancers-12-01032-t002:** Impact of CDSS on Process Outcomes.

Study	Installation	System Features	Key Outcomes Associated with CDSS	Results (Control vs. Intervention)
Adeboyeje et al., 2017 [26]	Web based	CPG ^1–4^ based, calculates febrile neutropenia risk and recommends CSF use	Percentage CSF use based on febrile neutropenia risk assessment	48.4% vs. 35.6%, *p* = 0.001, 95% CI: −14.7 to −2.7
Agiro et al., 2018 [27]	Web based	CPG ^3,4^ based, calculates febrile neutropenia risk and recommends CSF use	Percentage CSF use based on febrile neutropenia risk assessment	74.9% vs. 68.5%, *p* = 0.006, 95% CI: −6.0 to −4.7
Christ et al., 2018 [28]	Integrated in EMR	CPG ^3^ based, identifies patients who require pain assessment, displays patient-specific information and the most recent and maximum pain score	(a) Attainment of analgesia (defined as a pain score ≤ 4) at 24 h from admission (b) Time to analgesia (hours) (c) Percentage of documented pharmacy intervention (d) Mean frequency of pain assessments in first 24 h	(a) 33.3% vs. 43.8%, *p* = 0.78 (b) 14 vs. 15.9, *p* = 0.59 (c) Within first 24 h: 17.2% vs. 12.5%, *p* = 0.65. During entire admission: 31.0% vs. 56.3%, *p* = 0.32 (d) 7.4 vs. 12.0, *p* < 0.001
Verberne et al., 2012 [31]	Intranet-based	Assigns patients to one of three follow-up intervals based on CEA change, writes appropriate follow-up letter	(a) Working hour’s clinician for 5 years follow-up of a 200 patient cohort (b) Median follow-up in years for CRC patients after completion of treatment (c) Number of outpatient clinical visits to the surgeon for follow-up	(a) 1067 vs. 380 (b) 3.21 ((5% CI: 0.1 to 6.05) vs. 2.66 (95% CI: 0.2 to 10.8), *p* = 0.35 (c) 0 (95% CI: 0 to 7) vs. 3 (95% CI: 0 to 10), *p* < 0.001
Bouaud et al., 2001 [32]	Not mentioned	Hypertextual navigation in CPG * structured decision tree flowchart	(a) Percentage change between initial and final MDT treatment decisions (b) Percentage of patients recruited for clinical trials (initial vs. final)	(a) 31% (39/127) ** (b) 6.3% vs. 9.4%, z = 1.13
Jackman et al., 2017 [33]	Web based	Algorithms define the best fitting care pathway for patients at each point in care	Costs of care for 1 year after time of diagnosis in US dollar: (a) Adjusted costs (b) Unadjusted costs	(a) $69,122 (95% CI: 33,242 to 105,001) vs. $52,037 (95% CI: 25,200 to 48,849), *p* = 0.01 (b) $64,508 (95% CI: 53,140 to 75,876) vs. $48,515 (95% CI: 41,421 to 55,608), *p* = 0.03

CPG = clinical practice guidelines, CSF = colony stimulating factor, MDT = multidisciplinary team, CEA = blood biomarker Carcino-Embryonic Antigen, CRC = colorectal cancer, ^1^ = ESMO clinical practice guidelines, ^2^ = EORTC guidelines, ^3^ = NCCN guideline, ^4^ = ASCO guideline, * Unknown origin, ** This outcome represents the comparison of initial and final therapeutic decisions.

**Table 3 cancers-12-01032-t003:** Impact of CDSS on Guideline Adherence.

Study	Installation	System Features	Key Outcomes Associated with CDSS	Results (Control vs. Intervention)
Christ et al., 2018 [28]	Integrated in EMR	CPG ^1^ based, identifies patients who require pain assessment, displays patient-specific information and the most recent and maximum pain score	Percentage of guideline-adherent pain regimens	40.0% vs. 46.9%, *p* = 0.97
Rios et al., 2003 [29]	Not mentioned	CPG ^2^ based, organizes patient data and generates patient specific recommendations	Percentage of guideline-adherent treatment decisions (a) Breast cancer (b) Prostate cancer	(a) 77.8% vs. 87.1%, *p* = 0.02 (b) 86.7% vs. 89.9%, *p* = 0.35
Seroussi et al., 2007 [30]	Not mentioned	CPG ^3^ based, contextualizes both guideline medical knowledge and patient information and generates patient specific recommendations	Percentage of guideline-adherent treatment decisions	79.2% vs. 93.4%, *p* < 10^−5^
Bouaud et al., 2001 [32]	Not mentioned	Hyper textual navigation in CPG * structured decision tree flowchart	Percentage of guideline-adherent treatment decisions	61.42% vs. 85.03%, *p* < 10^−4^
Bertsche et al., 2009 [34]	Integrated in hospital drug information system.	Algorithms based on CPG ^4^ generate pain specific recommendations	Percentages of deviations from guidelines (a) On hospital admission (b) At discharge from hospital	Percentages of deviations (a) 80% vs. 85%, *p* = 0.6 (b) 74% vs. 14%, *p* < 0.001

CPG = clinical practice guidelines, EMR = electronic medical record, ^1^ = NCCN guideline, ^2^ = Oncolor guideline, ^3^ = CancerEst, ^4^ = WHO principles for pain therapy, * Unknown origin.

**Table 4 cancers-12-01032-t004:** Impact of CDSS on Clinical Outcomes.

Study	Installation	System Features	Key Outcomes Associated with CDSS	Results (Control vs. Intervention)
Adeboyeje et al., 2017 [26]	Web based	CPG ^1–4^ based, calculates febrile neutropenia risk and recommends CSF use	Percentage of patients at high risk for febrile neutropenia	2.8% vs. 4.3%, *p* = 0.927, 95% CI: −0.35 to 0.10
Agiro et al., 2018 [27]	Web based	CPG ^3,4^ based, calculates febrile neutropenia risk and recommends CSF use	Percentage of patients at high risk for febrile neutropenia	5% vs. 5.5%, *p* = 0.778, 95% CI: −0.2 to 0.3
Christ et al., 2018 [28]	Integrated in EMR	CPG ^3^ based, identifies patients who require pain assessment, displays patient-specific information and the most recent and maximum pain score	(a) Mean pain score (NVAS) at hospital admission (b) Mean pain score (NVAS) over the first 28 h	(a) 6.3 vs. 7.4, *p* = 0.063 (b) 4.9 vs. 4.2, *p* = 0.11
Verberne et al., 2012 [31]	Intranet-based	Assigns patients to one of three follow-up intervals based on CEA change, writes appropriate follow-up letter	(a) Percentage of metastases found in follow-up (b) Percentage of curative metastasectomy for metastases found in follow-up	(a) 13% vs. 9%, *p* = 0.06 (b) 1.6% vs. 3.8%, *p* = 0.13
Jackman et al., 2017 [33]	Web based	Algorithms define the best fitting care pathway for patients at each point in care	(c) Median overall survival in months	(c) 10.7 vs. 11.2 (n = 210), *p* = 0.08
Bertsche et al., 2009 [34]	Integrated in hospital drug information system	CPG^5^ based algorithms generate pain specific recommendations	Pain intensity score (NVAS) on day 5 after admission (a) At rest (b) During physical activity	(a) 2.4 vs. 2.0, *p* = 0.43 (b) 4.0 vs. 4.0, *p* = 0.89

CPG = clinical practice guidelines, CSF = colony stimulating factor, CDSS = computerized clinical decision support system, NVAS = numeric visual analogue scale, CEA = blood biomarker Carcino-Embryonic Antigen, ^1^ = ESMO clinical practice guideline, ^2^ = EORTC guideline, ^3^ = NCCN guideline, ^4^ = ASCO guideline, ^5^ = WHO principles for pain therapy.

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
