# Peer review of "The Effect of Higher Level Computerized Clinical Decision Support Systems on Oncology Care: A Systematic Review"

_cancers, 2020, doi:10.3390/cancers12041032_

Round 1
Reviewer 1 Report
Klarenbeek et al. present a systematic review on the effect of decision support systems in oncology. The rationale to perform this study is given. The methods are adequate. The results are well presented and the discussion is balanced.
I have some minor concerns to be addressed:
Abstract: Please mention already here which populations are addressed in the nine studies and that this systematic review is relevant above all for breast and lung cancer patients.
Please include a paragrph “future research” and tell us based on your systematic review which form of “Higher Level Computerized Clinical Decision Support Systems” are needed and how these systems should be prospectively assessed.
What is your opinion about the lack of improvement of clinical outcomes despite higher adherence to the guidelines? Can one conclude that the guidelines itself do not lead to better clinical outcomes?
Reviewer 2 Report
First of all, the authors employed PRISMA guideline but not PRISMA-P 2015 that is up to date and contains 17 check lists. Why didn't you use PRISMA-P 2015? The authors need to elaborate in the discussion section on the differences between PRISMA and PRISMA-P and possible differences in results.
From line 234 to 238, the authors mentioned that “Proper use of CDSSs is of major importance to ensure patient safety and to evaluate the effect of a CDSS. Bouaud et al. concluded that it was better not to use the CDSS than to use it improperly, as their results suggested improper use caused lower decision quality than not using the system.” This is very important. What is proper use of CDSS and improper use of CDSS? The authors need to describe them with specific example.
The paragraph of quality of evidence and the paragraph of strength and limitation partially contain overlap, so the continuous paragraphs are easier to read.
